# The Effect of Moderate-Intensity Physical Exercise on Some Serum Inflammation Markers and the Immune System in Rats Fed Intermittent Fasting with a High-Fat Diet

**DOI:** 10.3390/medicina59091687

**Published:** 2023-09-20

**Authors:** Nizamettin Günbatar, Bahattin Bulduk, Selver Bezgin, Gökhan Oto, Fahri Bayıroğlu, Mehmet Bulduk

**Affiliations:** 1Van School of Health, Van YuzuncuYıl University, 65090 Van, Turkey; bahattinbulduk@yyu.edu.tr (B.B.); selverbezgin@yyu.edu.tr (S.B.); mehmetbulduk@yyu.edu.tr (M.B.); 2Department of Pharmacology, Van YuzuncuYıl University, 65090 Van, Turkey; gokhanoto@yyu.edu.tr; 3Faculty of Medicine, Department of Physiology, Yıldırım Beyazıt University, 06200 Ankara, Turkey; fbayiroglu@aybu.edu.tr

**Keywords:** highfat diet, exercise, monocytes, intermittent fasting

## Abstract

*Background and Objectives*: This study aimed to investigate the impact of moderate-intensity physical exercise on serum inflammation markers and the immune system in rats that were fed a high-fat diet (HFD) with intermittent fasting. *Materials and Methods*: A total of 48 Wistar albino male rats were included in the study and divided into eight groups, each consisting of six rats. Group 1 served as the control group (CG), receiving a standard diet. Group 2 followed the standard nutrition program with intermittent fasting (CG + IF). Group 3 underwent exercise with a standard diet (CG + E). Group 4 underwent both a standard diet with intermittent fasting and exercise (CG + IF + E). Group 5 was fed a high-fat diet (HFD). Group 6 received a high-fat diet with intermittent fasting (HFD + IF). Group 7 followed a high-fat diet with exercise (HFD + E). Group 8 underwent both a high-fat diet with intermittent fasting and exercise (HFD + IF + E). The study lasted for 8 weeks. *Results*: The results of the analysis show that lymphocyte cell levels in groups HFD + IF, HFD + IF, and HFD + IF + E were higher compared to groups CG-HFD (*p* < 0.05). Additionally, B lymphocyte and monocyte cell levels were higher in group HFD + IF + E compared to groups CG, CG + IF, and CG + IF + E, as well as CG, CG + IF, and CG + E, respectively. TNF-α levels were significantly higher in group HFD compared to the other groups. Furthermore, IL 10 levels were higher in group HFD + IF + E compared to the other groups. *Conclusions*: These findings indicate that moderate exercise and intermittent fasting, particularly in groups fed a high-fat diet, increased anti-inflammatory cytokine levels, and certain immune system cell counts, while decreasing pro-inflammatory cytokine levels.

## 1. Introduction

High-fat diets cause disruption of energy balance and increase in adipose tissue mass. Exercise plays an important role in the development of human health and protection from diseases by increasing energy consumption to create healthy adipose tissue, reduce metabolic risk factors, and prevent obesity [1]. A chronic low-grade inflammatory condition characterized by high cytokine levels is usually found in the obese and inactive population, and this condition is associated with cardiovascular and metabolic diseases [2]. It is reported that exercise can provide good protection against the development of these chronic diseases by offering a non-pharmacological intervention effect due to its anti-inflammatory effect. Moderate exercise leads to lower retention of circulating pro-inflammatory cytokines and increased expression of anti-inflammatory cytokines [3].

It is widely acknowledged that HFDs and obesity are the triggers and initiators of numerous diseases [4]. It was shown that HFDs induce alterations in gut bacterial composition [5]. Recent research focused on the potential therapeutic effects of various dietary therapies, including intermittent fasting, calorie restriction, and protein restriction. An increasing number of studies were conducted on the positive effects of intermittent fasting and fasting imitation diets in terms of delaying aging, improving quality of life, and preventing various pathologies associated with aging [6,7,8,9].

Intermittent feeding has positive effects on reducing inflammation and increasing the ability of macrophages to kill against mycobacterium tuberculosis [10]. Intermittent feeding leads to an increase in B cell immunity [11], plasma interferon-ϒ [12], and especially the activity of white blood cells such as neutrophils, natural killer cells, and monocytes cells [13].

Regular physical exercise also has positive effects on the prevention of cardiac-related death, obesity, osteoporosis, and hypertension [14]. Furthermore, exercise-induced changes in immune parameters recently became a major concern among researchers and also stimulated research into the effect of exercise on physical fitness and immune response [15]. Light and moderate intensity exercise was shown to strengthen the immune system, reduce the risk of respiratory infections and buffer against stress-related disorders such as anxiety and depression [16]. In contrast, irregular prolonged high-intensity exercise was shown to impair the immune system and increase the risk of infectious and allergic diseases [17]. For these reasons, regular moderate exercise programs are commonly recommended and practiced to strengthen the immune functions in patients with chronic diseases [1]. Exercise increases the levels of proinflammatory cytokines (such as IL-6, IL-1β; TNF-α) that mediate the communication between immune and non-immune cells to initiate repair processes in the organism [18] and this is accompanied by the release of anti-inflammatory cytokines that have the function of reducing the inflammatory process [19,20].

In this study, we aimed to investigate the effect of moderate-intensity physical exercise on some serum inflammation markers and the immune system in rats fed a high-fat diet (HFD) with intermittent feeding.

## 2. Materials and Methods

Prior to the study, the study was approved by Van Yuzuncu Yil University Rectorate Animal Experiments Local Ethics Committee with Decision 2015-08. A total of 48 male Wistar Albino rats were used for the experiment. All the procedures except for the experimental diet were performed in accordance with the standard guidelines for the care and use of laboratory animals [21]. The rats were divided into 8 groups with 6 rats each: I control group were fed a standard diet (2.8% crude fat, 23.1% crude protein, 5% crude cellulose, 7.1% crude ash, and 12.8% moisture), group II were fed a standard diet and underwent intermittent fasting (a 24 h severe restriction excluding water for 2 non-consecutive days a week), group III were fed a standard diet and underwent exercise training (3 days a week), group IV were fed a standard diet and underwent intermittent fasting (2 days/week) and exercise training (3 days/week), group V were fed a HFD (standard pellet chow added with 300 g/kg melted margarine, prepared daily and administered for 8 weeks), ref. [21] group VI were fed a HFD and underwent intermittent fasting, group VII were fed a HFD (with 60% of the energy from saturated fat) and underwent exercise training, group VIII were fed with HFD and underwent intermittent fasting and exercise training. The study was conducted over a period of 8 weeks [22]. At the end of this period, preanesthesia was induced with ketamine + diazepam (Sigma K2753) (50 + 8 mg/kg) andblood samples were taken from the heart of the rats and were transferred into biochemistry tubes. The tubes were centrifuged at 4000 RPM (RCF = 1240× *g*) for 15 min and then serum was separated. While T-helper cells, natural killer (NK) cells, and B-lymphocytes in whole blood were measured using a BD FACS CANTO II device with the flow cytometry method at Van Yuzuncu Yil University Dursun Odabası Medical School Microbiology Laboratory, neutrophil, eosinophil, basophil, and monocyte cell levels in whole blood were measured using a Vega group MS4-S device with the laser method at Van Yuzuncu Yil University Veterinary School Animal Hospital Internal Diseases Hematology Laboratory. IL-1β, IL-4, IL-10, IL-12, IGF-1 and TNF-α (Sunred Elisa Rat kit, catalogue no. 201-11-0120, 201-11-0134, 211-11-0109, 201-110112, 201-11-0710, and 201-11-0765) levels were measured by the double-antibody sandwich ELISA method in Van Yuzuncu Yil University Dursun Odabaş Medical Center hematology laboratory.

### 2.1. Flow Cytometry

After pipetting 100 µL (microliters) of blood into a 12 × 75 falcon tube, one of the antibodies to be used was CD4 Thc, 20 µL from CD56 PE, and 5 µL from CD19 APC were pipetted into a tube and vortexed, and then incubated in a dark environment and at room temperature for 15–30 min. After incubation, 1.5–2 mL of lysing buffer was added and vortexed and incubated in the dark for 8–10 min and centrifuged at 2000 (rpm) for 5 min. After the supernatant was discarded, the pellet was vortexed, 2 mL of CellWash was added and vortexed again, and then centrifuged at 2000 (rpm) for 5 min. After the supernatant was discarded, the pellet was vortexed and 500 µL cell Wash was added to ensure vortexation and then it was read in the device. The java-based Diva software (bdbiosciences.com), which can be used for research cytometry or clinical samples, was used for flow cytometry data analysis. Table 1 shows the antibodies used in Flow Cytometry and the targeted cell types, conjugated fluorophores, columns and isotypes.

Anti-CD4, CD56, and CD19 antibodies were used to detect T helper, natural killer, and B lymphocyte cells. The T cells were found to be CD4+, while the natural killer cells were CD56+ and the B lymphocytes were CD19+. The quantitative values of these cells were determined using the erythrocyte lysis and density gradient centrifugation method.

### 2.2. Exercise Training

For the exercise training, the rat groups underwent training on a treadmill. During the 2-week pretraining period, the rats ran at the slowest speed specified in the protocol, which involved 15 min per day at a speed of 17 cm/s. However, during the 8-week exercise training, the rats ran at a moderate intensity for 30 min per day at a speed of 40 cm/s, following the treadmill protocol developed by Rico et al. [23].

### 2.3. Statistical Analysis

Statistical analysis was conducted by calculating the sample size to achieve at least 80% power for each variable, assuming a Type-1 error of 5%. The normality of the measurement distributions was assessed using the Shapiro–Wilk test (for sample sizes less than 50). Parametric tests, such as unidirectional ANOVA, were used for measurements with normal distributions, while nonparametric tests, such as the Kruskal–Wallis test, were used for measurements without normal distributions. Descriptive statistics, including mean, standard deviation, median, minimum, and maximum, were used to summarize continuous variables in the study. The statistical significance level was set at 5%, and the analysis was conducted using the SPSS statistical package program, version 26, by IBM.

## 3. Results

Table 2 shows a significant increase in lymphocyte/mm^3^ levels in the HFD + IF, HFD + E, and HFD + IF + Eth groups compared to the CG-HFD group (*p* < 0.05). Moreover, the WBC/mm^3^ levels in the HFD + IF, HFD + E, and HFD + IF + E groups were significantly higher than the CG + IF group (*p* < 0.05). Additionally, T-helper cells levels were higher in the HFD + IF, HFD + E, and HFD + IF + E groups compared to the CG + IF, CG + IF + E groups (*p* < 0.05).

Furthermore, Table 3 demonstrates that the natural killer cells level was higher in the HFD group compared to the other groups (*p* < 0.05), while the B lymphocyte and monocyte cells levels in the HFD + IF + E group were higher than in the CG, CG + IF, CG + IF + E, CG, CG + IF, and CG + E groups (*p* < 0.05).

Moreover, Table 4 reveals that the TNF-α level was higher in the HFD group compared to the other groups (*p* < 0.05) (Table 3, Figure 2D). On the other hand, the IL-10 level was higher in the HFD + IF + E group compared to the other groups (*p* < 0.05).

The lymphocyte cells levels in the groups HFD + IF, HFD + E, HFD + IF + E was higher than the groups CG-HFD (*p* < 0.05) (Table 2, Figure 1B), while the WBC cell levels in the groups HFD + IF, HFD + E, HFD + IF + E were higher than the group CG + IF (*p* < 0.05) (Table 2, Figure 1D). In addition, the T-helper cell levels were higher in the groups HFD + IF, HFD + E, HFD + IF + E compared with the groups CG + IF, CG + IF + E (*p* < 0.05) (Table 2, Figure 2A), while the B lymphocyte and monoctyes cell levels in the group HFD + IF + E were higher than in the groups CG, CG + IF, CG + IF + E and CG, CG + IF, CG + E (*p* < 0.05) (Table 3, Figure 1C), (Table 3, Figure 2B). The naturel killer cells level was higher in the group HFD compared with the other groups (*p* < 0.05) (Table 3, Figure 1A), while the TNF-α level was higher in the group HFD compared with the other groups (*p* < 0.05) (Table 4, Figure 2D). However, the IL 10 levels were higher in the groups HFD + IF + E compared with the other groups (*p* < 0.05) (Table 4, Figure 2C). IGF-1,IL-4, IL-12, neutrophils, eosinophils, basophils, IL-1β levels were not statistically significant (Table 2, Table 3 and Table 4, Figure 3A–D and Figure 4A–C). Figure 5 shows the Flow cytometry gating procedure.

## 4. Discussion

The metabolic and immune systems are closely interrelated, and their functions are interdependent. Excessive food intake, leading to obesity, can be considered a stress-related biological event that is harmful to the body. This activates inflammation and stress responses in various metabolic tissues, causing a low-grade chronic inflammation known as metabolic inflammation or meta-inflammation. Inflammation is a necessary physiological response to restore homeostasis when disrupted by stimuli. However, prolonged inflammation or an excessive response can have detrimental effects. Individuals who are overweight or obese often experience a persistent and low-grade inflammatory state. During obesity, circulating levels of pro-inflammatory cytokines are elevated compared to those in individuals of normal weight, and these cytokines are believed to contribute to the development of insulin resistance. Although adipose tissue is the main source of these cytokines, infiltrating macrophages also play a significant role. Thus, blood concentrations of these cytokines decrease after weight loss [24].

Exercise is known to have positive effects on the immune system and is considered one of the best therapy methods for both cancer patients and healthy individuals [25]. Physical exercise, regardless of short- or long-term application, was shown to increase leukocyte cell count. However, long-term intensive exercise was found to decrease the total leukocyte cells count [26]. Furthermore, acute exercise and hypoxia were reported to induce changes in the levels of circulating lymphocytes and neutrophils cells [27]. Exercise also affects neutrophils function in both the short and long term. Acute exercise results in a rapid increase in blood neutrophil cell count within a few minutes, followed by a decrease below the normal level within several hours [28,29].

White blood cells, or leukocytes, are the main cells of the immune system and are produced from hematopoietic stem cells in the bone marrow through a developmental process called hematopoiesis, where the cells are divided into erythroid, myeloid, and lymphoid lineages [30]. Myeloid cells can be classified into two groups; granulocytes containing reactive substances that can destroy the microorganism and play an important role in inflammatory events. Of these, neutrophils are the most abundant granulocytes, followed by eosinophils and basophils. The second group of myeloid cells includes monocytes, macrophages, and dendritic cells. Monocytes are leukocytes that circulate in the blood and are mobile parts of dormant tissue cells called macrophages whose main function is phagocytosis [30]. Lymphocytes are defined according to their morphology; natural killer cells (NK cells) are large lymphocytes that are effector cells of innate immunity and their primary function is defense against viral infections. They prevent the spread of infection by killing virus-infected cells and releasing cytokines that block viral replication. Small lymphocytes, including B and T cells, are responsible for adaptive immune responses [31].

Dietary interventions such as intermittent fasting and time-restricted feeding showed promise in preventing and treating certain chronic metabolic diseases while restoring the balance between nutrition and fasting. In particular, a study by Cisse et al. (2018) demonstrated that a time-restricted diet could alter the development of innate immune responses in mice [32].

Previous studies reported varying effects of exercise on different immune cell populations. McCarthly and Dale (1988) reported an increase in neutrophil count during and after long-term exercise training [33]. Similarly, Brines et al. observed an increase in levels of neutrophils, monocytes, and lymphocytes [34]. Conversely, Pedersen and Nieman (1998) reported a decrease in lymphocyte, CD19+, CD16 + 56+, and NK cell levels after exercise [35]. Another study indicated that monocyte count increased in the exercise group compared to the control group after a 12-week endurance exercise performed twice a week [36]. In contrast, Woods et al. (1999) found no significant change in neutrophil and monocyte counts after a 6-month moderate-intensity exercise training [37]. In an acute intermittent exercise study conducted by Evans et al., a significant increase in total lymphocyte cells values was observed after exercise, along with an insignificant increase in monocytes, leukocytes, and NK cells values [38]. Ladha et al. reported an increase in neutrophil cells counts in their treadmill exercise study [39].

On the other hand, the available literature contains limited documentation regarding the effects of intermittent fasting on the immune system. Intermittent fasting during Ramadan causes significant decreases in circulating immune cells and proinflammatory cytokines [40,41]. 

In a study conducted on 50 volunteers by Faris et al., it was reported that the total leukocyte and lymphocyte values of immune system cells decreased, while the monocyte cells values increased significantly [40]. Similarly, Latifynia et al. (2009) found that the Ramadan fasting had a beneficial effect on the phagocytic function of neutrophils cells [41]. Nevertheless, Nasiri et al. (2017) evaluated 28 healthy subjects and reported that the neutrophil and lymphocyte counts decreased at the fourth week of Ramadan and at three months after Ramadan [42]. Moro et al. (2020) found that there was no change in eosinophil basophil, monocyte cells, tnf-α, ıgf-1, and leukocyte cell values in a time-restricted intermittent feeding study, while neutrophil cell values decreased significantly and lymphocyte values increased significantly [43]. Faietti stated in his study that there was a significant decrease in lymphocyte cell levels in the exercise group in which intermittent nutrition was applied, an increase in granulocyte, no difference was observed in monocytes cells count, and the increase in t- helper (cd4) cells was insignificant [44]. Faris et al. (2012) reported that there was a dramatic decrease in lymphocyte cell count and tnf-α values and inflammation during Ramadan compared to the period before Ramadan [40].

In this study, we found that monocytes cell levels (0.46/mm^3^) in the HFD + IF + E group were significantly higher compared to the CG, CG + IF, CG + Egroups (*p* < 0.05) (Table 3, Figure 2B), while the differences in neutrophils, eosinophils, and basophils cell count values between the groups were not found to be significant (Table 2 and Table 3, Figure 3D and Figure 4A,B). In addition, the WBC cells count levels in the HFD + IF, HFD + E, HFD + IF + E groups was higher 93.53% compared to the CG + IF group (*p* < 0.05) (Table 2, Figure 1D).

Previous studies showed that the most significant effect of physical exercise on immune parameters is the increase in neutrophil and monocyte cell counts noted after long-term exercise training [33]. This increase was shown to be associated with the intensity and duration of the exercise training [45]. In our study, we think that the increased monocyte cell counts in the HFD + IF + E group (Table 3, Figure 2B) groups, especially in the exercise program applied, are related to the duration and intensity of the exercise program that we applied.

Natural killer (NK) cells were first identified as major tissue fitness complex unlimited killers in the early 1970s due to their ability to spontaneously kill certain tumor cells [46].

Natural killer cells (NK), which have a direct effect and the ability to kill, especially on virally infected cells, as well as cells that underwent malignant transformation, are an important component of the immune system [46]. They typically follow a profile that shows significant increases in both cell count and activity immediately after exercise, with decreases for up to several hours during the recovery period [16]. It is also thought that exercise-related increases in NK cell function may provide a protective effect. Exercise-induced decreases in NK cell function can lead to an increase in the incidence of viral infection and potential disease [47].

Moderate-to-high-intensity resistance and aerobic exercise have the potential to maintain NK cell function. Natural killer (NK) cells activity is known to increase either during or within several minutes after moderate and high-intensity exercise training. However, the intensity of exercise training, rather than its duration, is considered to be more responsible for the increase in NK cells activity [48]. There are various biological mechanisms associated with the increase and decrease in NK cell numbers during and after moderate-intensity acute aerobic exercise. Increased cardiac output during exercise may have a synergistic effect with increased catecholamine levels, which provide vascular endothelial lymphocyte adhesion that will attract NK cells from NK cells reservoirs. The decrease in NK cell numbers during the recovery period after exercise is associated with changes in exercise-induced catecholamineand cortisol levels, and these changes are mediated by changes in the inflammatory cytokine interleukins [25,49].

Niemen et al. (1990) stated that the number of natural killer cells and natural killer cell activity were observed as the number of natural killer cells (0.256, 0.264, and 0.241) and natural killer cells activity (36.2, 55.7, and 51.9) in exercise studies of 45 min, 5 days a week. It can be said that the number of cells has an effect on cell activity with this result, which they realize in the 6-week and 15-week comparisons, respectively [50]. Kendall et al. (1990) and Nieman et al. (1993) reported that NK cell activity increased due to increased NK cell numbers due to exercise and increased cell numbers [51,52]. In our study, it was seen that the number of natural killer cells (7.02 CD56/mm^3^) in the fifth group fed a high-fat diet (Table 3) tended to increase compared to all other groups, and this increase was statistically significant (*p* < 0.05). A high-fat diet was reported to increase natural killer cell production [53]. In our study, we think that the reason for the increase in the number of NK cells in the HFD group is due to the application of a high-fat diet.

Lymphocytes are defined according to their morphology; natural killer cells (NK cells) are large lymphocytes that are effector cells of innate immunity and their primary function is defense against viral infections. They prevent the spread of infection by killing virus-infected cells and releasing cytokines that block viral replication. Small lymphocytes, including B and T cells, are responsible for adaptive immune responses [31]. 

Literature presents contradictory results regarding the lymphocyte counts measured after physical exercise. Although some studies showed reduced lymphocyte [54], some others showed elevated counts [55,56]. Additionally, some other studies indicated that no change was observed in the lymphocyte cell counts after physical exercise [57]. Shimizu et al. (2011) reported that the no change was observed in the counts of lymphocytes and T-helper (CD4) cells in the exercise group compared to the control group after an endurance exercise performed 2 days/week for 12 weeks [35]. Similarly, Campbell et al. (2008) also reported that no change was found in the counts of T-helper and B-lymphocytes cells in the exercise group compared to the control group after an aerobic exercise performed in 30 min sessions 2 days/week for 12 months [58]. In another study, Sitlinger et al. reported that CD4+ T cell counts increased after exercise in the treadmill exercise study [59]. Campbell and Turner reported that the total number of B cells increased after exercise [60].

Natale et al. (2003), in three different exercise studies they conducted, showed that lymphocyte cell counts increased significantly in three exercise groups, CD + 4 (T-helper) cell count increased significantly in the peak aerobic and prolonged exercise group, and the increase in the resistance exercise group was not significant. It was reported that the number of B lymphocyte (CD + 19) cells increased immediately after exercise in the peak aerobic exercise group [61].

Moreover, Woods et al. (1999) also reported that no change was observed in the counts of T-helper and lymphocytes cells in the exercise group after a 6-month moderate-intensity exercise training [37]. In our study, the lymphocyte cell values in the HFD + IF, HFD + E, HFD + IF + E groups were higher compared to the CG-HFD groups (*p* < 0.05) (Table 2, Figure 1B), while the T-helper levels were higher in the HFD + IF, HFD + E, HFD + IF + E groups compared with the CG + IF, CG + IF + E groups (*p* < 0.05) (Table 2, Figure 2A), the B lymphocyte cells levels in the group HFD + IF + E were higher than in the groups CG, CG + IF, CG + IF + E (Table 3, Figure 1C) (*p* < 0.05).

Although IGF-1 mostly originates in the liver, it can also be produced locally by most tissues, and thus it can act in an endocrine manner while also being active in an autocrine/paracrine manner [62]. The presence of high levels of IGF-1 in the circulation for a long time was stated to cause many carcinogenic stages to progress and develop cancer due to the fact that IGF-1 has mitogenic and antiapoptic effects [63].

It was stated that after twice weekly resistance exercise for 15 weeks with 43 women (16 were underweight and 17 were overweight), the level decreased in underweight women, but this decrease was not observed in overweight women [64]. Valkeinen et al. (2005) reported that resistance exercise did not change the IGF-1 level [65].

In a high-intensity exercise study on 39 sedentary and athletic men, Herbert et al. found that IGF-1 values were higher in athletic men than in sedentary ones [66]. It was reported that serum IGF-1 levels increased after moderate swimming exercise by Gomes et al. [67]. Özbeyli et al. (2017) observed that IGF-1 levels increased in all exercise types after a study consisting of three groups in which aerobics, resistance, and aerobic-resistance exercise applied 3 days a week for 6 weeks [68]. Dunn et al. (1997) found a decrease in serum IGF-1 levels in the group on which they applied dietary restriction [69]. In another study that practiced intermittent feeding, IGF-1 levels increased in the group that intermittent feeding was applied on while it was decreased in the group that was fed by 60% of daily nutrition [70].

In our study, although the difference in the value of IGF-1 levels between the groups was not significant, there was a non-significant increase in the group HFD + IF + E while it tended to decrease in the group with a high-fat diet and the group fed with a high-fat diet and an exercise program (Table 4, Figure 3A). TNF-α, which is responsible for the increase in atherosclerotic plaques, exerts this effect by decreasing the level of IGF-1 and increasing the level of IGFBP-3 [71]. In our study, the level of IGF-1, which tended to increase, especially in the group HFD + IF + E, is considered to be due to TNF-α, which decreased significantly in the group HFD + IF + E (Table 4, Figure 2D).

Interleukin 10 is a powerful anti–inflammatory cytokine and plays a central role in host defense against foreign pathogens. It has the property of ensuring the continuation of normal tissue balance and protecting the host from damage. IL-10 dysregulation is associated with an increased risk of developing many autoimmune diseases, as well as the development of immunopathology [72].

The main production sources of IL-10 are T-cells, monocyte, macrophage, and dendritic cells, as well as B cells from myriad immune effector cells, cytotoxic T cells, NK cells, mast cells, and granulocytessuch as neutrophils andeosinophils can produce IL-10 [73,74]. 

IL-10, which has the property of protecting the organism from systemic inflammation, is an important anti-inflammatory cytokine and regulates the levels of proinflammatory cytokines by decreasing them [75]. A study on rats found that the IL-10 level in the group thatdid strenuous exercise was higher significantly in (*p* < 0.05) than in the group that exercised moderately [76]. In their study of 10 marathon runners at moderate speed, Otrowski et al. reported a limited increase in the level of IL-10, which is an anti-inflammatory, after running [77]. It was also reported by Cabral-Santos et al. that the amount of interleukin-10 (IL-10), which has an anti-inflammatory role, increased in the circulation in response to exercise [78]. Another study reported no change in the IL-10 value after moderate exercise compared to sedentary individuals [79]. Helmark et al. (2010) reported that IL-10 values increased in the exercise group in their resistance exercise application [80].

In a study by Vasconcelos et al. (2014), intermittent feeding increased IL-10 levels [81], and the same conclusion was reached by Sharman and Volek (2004) as well as Ugochukwu et al. (2007) [82,83]. In our study, the IL-10 value was significantly *p* < 0.05 (402.237 pg/mL) higher in the group HFD + IF + E than the other groups (Table 4, Figure 2C). The main production sources of IL-10 are T-helper, monocyte, macrophage, and dendrite cells [72,73], and plasma increases IL-10 levels in the organism in the IGF-1 administration without the formation of pancreatitis [84]. In our study, the increased IL-10 level is considered due to IGF-1, which also increased at a level that was not significant in the same group (Table 4, Figure 2C).

Interleukin-1β (IL-1β) is a powerful proinflammatory cytokine and has an important role in host defense against injury and infection. Although there are many studies showing that it is produced from innate immune system cells such as monocytes and macrophages, it is also secreted and produced by various cells [85]. In a study, it was stated that IL-1β decreased significantly (*p* < 0.05) after Ramadan in a comparison before, during, and after Ramadan [40]. Sharman and Volek. (2004), Ugochukwu et al. (2007), and Harpal et al. (2017) reported that intermittent feeding reduces the value of IL-1β [82,83,86].

In a moderate exercise study, there was no change in IL-1 β values after moderate exercise compared to sedentary individuals [79]. In their study of 10 moderate-speed marathon runners, Ostrowski et al. (1999) found that IL-1β, which is a proinflammatory cytokine, increased after running [77]. IL-10 inhibits the production of IL-1β and TNF-α [87].In our study, there was no significant difference in the value of IL-1β levels between the groups (Table 4, Figure 4C), which is considered to be due to the significantly increased IL-10 (Table 4).

Tumor necrosis factor alpha TNF-α is an inflammatory cytokine produced by macrophages/monocytes during acute inflammation, and while it is also responsible for some of the signals that cause the onset of intracellular events that cause necrosis and apoptosis, it is important in the resistance to infection and cancer formation [88,89]. Serum TNF-α levels increase with inflammation, septic shock, rheumatoid arthritis, host parasitic diseases, obesity, and developing insulin resistance. Its main important effects are oxidative stress in endothelial cells, adhesion molecules, angiogenesis in endothelial cells, and increasing proliferation in t- helper (CD4+) cells, regulating hematopoiesis. The level of TNF-α increases in inflammatory diseases, viral and bacterial infections, while it decreases in autoimmune diseases [90,91].

Dimitrov et al. (2017) reported that there was no change in TNF-α values before and after exercise in a study with 47 volunteers [92]. Accattato et al. (2017) reported that there was no change in TNF-α values after walking exercise on a treadmill for 20 min 3 days a week with 30 volunteers [93]. Colbert et al. (2004) reported that TNF-α (*p* < 0.2) decreased significantly after high levels of exercise [94]. In another study, Jiménez-Maldonado et al. (2019) reported that TNF-α values in exercise groups were lower than control groups in moderate and high intensity exercise study [95]. In another moderate exercise study, Elosua et al. (2005) reported that TNF-α value decreased after moderate exercise in the exercise group compared to sedentary individuals [80].

Faris et al. (2012) found that the serum TNF-α value in the third week of Ramadan (52.22 ± 57.25 pg/mL) was significantly lower than the pre-Ramadan serum TNF-α value (179.62 ± 129.56 pg/mL) (*p* < 0.001) [40]. In the study of Park et al. (2012), while TNF-α levels were high in the group fed a high-fat diet, no increase in TNF-α was observed in the calorie-restricted group [96]. Vasconcelos et al. (2014) found that the increased Tnf-α’ value in rats administered lipopolysaccharide (LPS) was statistically decreased in the intermittent fasting (IF) + LPS group (*p* < 0.001) [81]. Harpal et al. (2017) reported that intermittent feeding decreased TNF-α levels [86]. Similar results areshown in two other studies [82,83].

IL-10 was reported to inhibit the production of TNF-α [87]. In the presented study, the TNF–α value in the groups HFD + IF + E (18.217 pg/mL) was significantly lower (*p* < 0.05) 22.18% than the group HFD (Table 4, Figure 2D), which is considered to be due to the IL-10 value that also increased in the same group (Table 4). 

IL-4 was discovered in the mid-1980s as a multifunctional pleotropic cytokine. While it is produced by activated T cells, it is also produced by mast cells, basophils, and eosinophils [97]. IL-4 plays a central role in the regulation of the innate immune response. Its main physiological effect is to regulate allergic events. They originate from antigen-stimulated CD4 + T lymphocytes, particularly the TH2 subgroup, and are responsible for the growth, activity, and differentiation of B and T lymphocytes [98].

Since anti-IL-4 serum regulates normal monokine production, IL-4 inhibits the secretion of TNF-alpha and IL-1-beta [99]. Accacato et al. (2017), in their study on 30 volunteers, reported that there was no change in IL-4 values after walking exercise on a treadmill for 20 min 3 days a week [93]. The same result was reported by Zamani et al. (2017) [100]. Conroy et al. (2016) found no significant change in IL-4 in a study of a moderate regular exercise program lasting 5 days a week for 6 months [101]. In the study of Abd El-Kader et al. (2016), in which 80 people performed treadmill aerobic exercise for 6 months, IL-4 level decreased significantly in the experimental group after exercise [102]. In another study, IL-4 increased after exercise in a treadmill study on rats [103].

In our study, no significant difference was found between the groups in IL-4 value (Table 4, Figure 3B). Iwaszko et al. (2021) stated that IL-4 was also produced by eosinophil and basophil cells [97]. This result in our study is thought to be due to the insignificant difference between the groups in the number of eosinophils and basophils cells that produce IL-4 (Table 2 and Table 3).

IL-12 is an important cytokine that coordinates the immune response necessary to cope with bacterial and viral infections invading the immune system. In particular, it plays an important role in the regulation of T-cell response. These responses are regulated by dendrite cells, macrophages, and monocytes, enabling the production of IL-12 in response to infection [104]. In their study, they found no change in IL-12 values after exercise [100]. Kohut et al. (2001) reported there was no change in IL-12 value after the study in an 8-week moderate treadmill exercise study in 4-month-old young mice [105]. The IL-12 value increased in sedentary rats after exercise in a 9-week treadmill exercise program on 120 rats [106]. IL-12 values decreased significantly in the fourth week of Ramadan and in the exercise group in their study during Ramadan [107]. It was reported that IL-10 inhibits the production of IL-12 [108]. In our study, the difference between the groups of IL-12 was not found to be significant (Table 4, Figure 3C), and this result is considered to be due to be due to the IL-10 level, which increased by *p* < 0.05 in all groups (Table 4).

## 5. Conclusions

As a result, according to our study results, moderate exercise and intermittent feeding practice, especially in groups fed with a high-fat diet, increased anti-inflammatory cytokines with some immune system cell counts, decreased pro-inflammatory cytokines were observed, and considering that this study will guide studies in this area, it is recommended to conduct more advanced studies, especially on immune system cell activities, using a similar method.

## Figures and Tables

**Figure 1 medicina-59-01687-f001:**
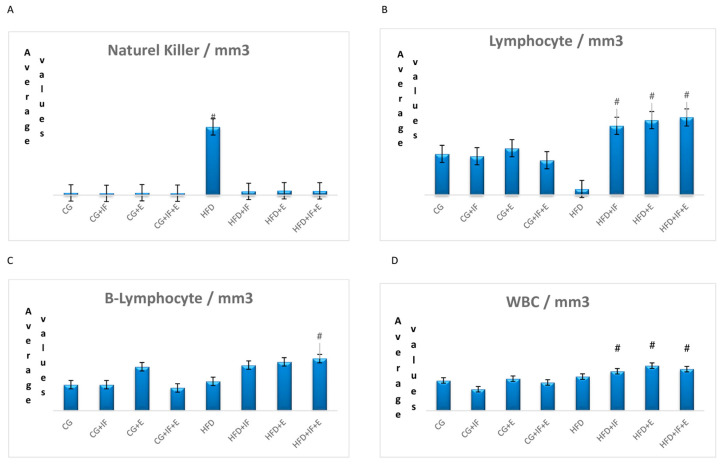
The mean values of, naturel killer (**A**), ymphocyte (**B**), B-lymphocyte (**C**), and WBC (**D**) among the groups. # *p*: significant (*p* < 0.05).

**Figure 2 medicina-59-01687-f002:**
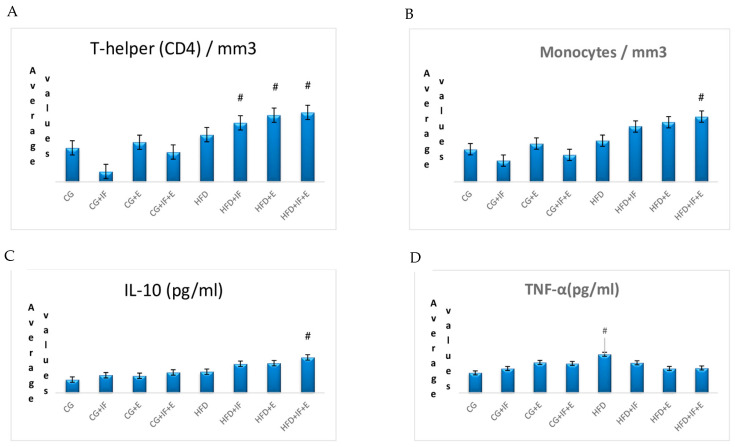
The mean values of T–helper (**A**), monocyte (**B**), IL-10 (**C**), TNF-α,and (**D**) among the groups. # *p*: significant (*p* < 0.05).

**Figure 3 medicina-59-01687-f003:**
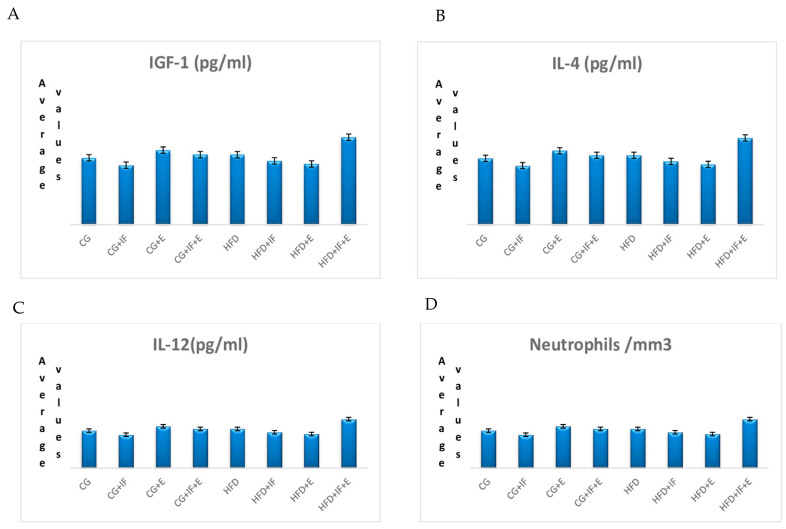
The mean values of IGF-1 (**A**), IL 4 (**B**), IL-12 (**C**), neutrophils, and (**D**) among the groups.

**Figure 4 medicina-59-01687-f004:**
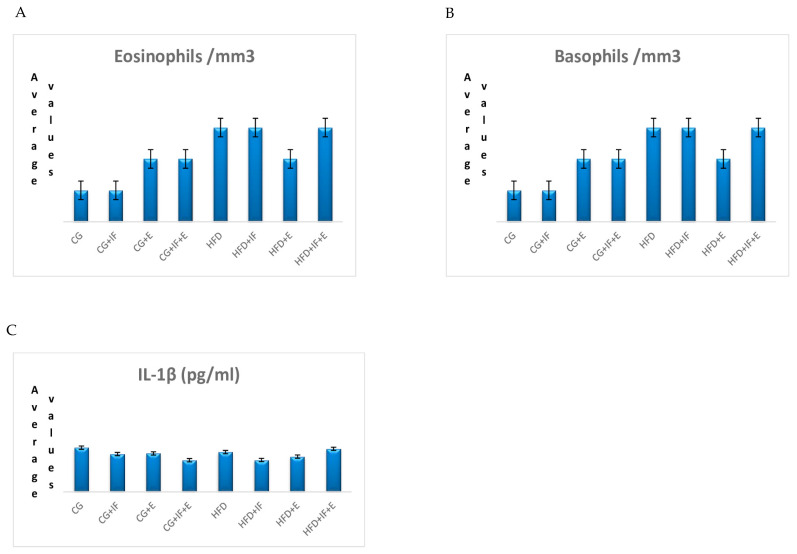
The mean values of eosinophils (**A**), basophils (**B**), IL-1β, and (**C**) among the groups.

**Figure 5 medicina-59-01687-f005:**
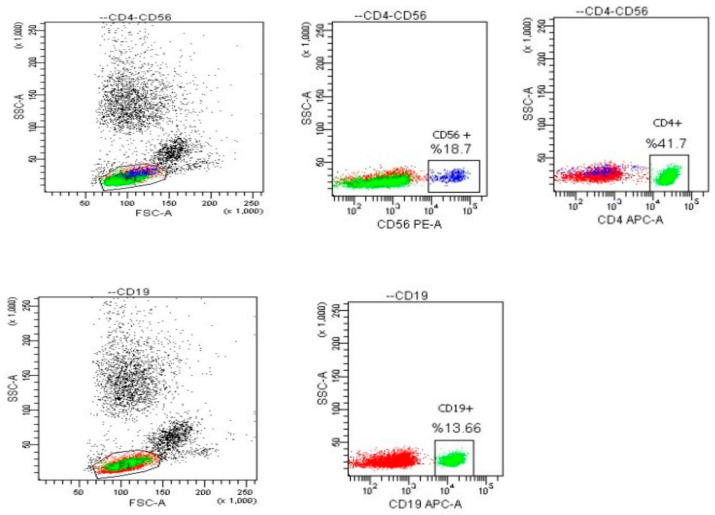
Flow cytometry gating procedure.

**Table 1 medicina-59-01687-t001:** The antibodies used and the targeted cell types, conjugated fluorophores, columns, and isotypes were shown.

Antibody	Cell Type	Fluorophore	Clone	Isotype
Anti-rat CD4 antibody	T cell	FITC	RM4-5	Mouse IgG2a K
Anti-rat CD56 antibody	NK cell	PE	NCAM-1	Mouse IgG1 K
Anti-rat CD19 antibody	B cell	APC	SJ25C1	Mouse IgG1 K

**Table 2 medicina-59-01687-t002:** Mean values counts of WBC, neutrophils, basophils, lymphocyte, and T-helper cells in all groups.

	Group	$\bar{X}\;\pm \;Sx$ Median (Min–Max)	*p*
WBC/mm^3^	CG	7.41 (6.16–9.35) ab	0.003 *
	CG + IF	5.26 (0.00–8.28) b	
	CG + E	7.85 (6.26–10.65) ab	
	CG + IF + E	6.92 (4.50–9.05) ab	
	HFD	8.36 (5.40–13.27) ab	
	HFD + IF	9.67 (6.86–13.01) a	
	HFD + E	11.04 (8.35–14.45) a	
	HFD + IF + E	10.18 (6.85–11.82) a	
	CG	1.23 (0.98–1.45) c	0.001 *
	CG + IF	1.16 (0.42–2.15) c	
	CG + E	1.40 (0.95–1.96) c	
Lymphocyte/mm^3^	CG + IF + E	1.04 (0.60–1.48) c	
	HFD	0.18 (0.10–0.41) d	
	HFD + IF	2.07 (1.48–2.93) ab	
	HFD + E	2.24 (1.39–2.78) a	
	HFD + IF + E	2.32 (1.63–2.72) a	
Neutrophils/mm^3^	CG	3.60 (2.64–5.58)	0.115
	CG + IF	2.81 (0.42–4.30)	
	CG + E	3.77 (2.19–5.50)	
	CG + IF + E	3.64 (2.21–4.86)	
	HFD	3.98 (2.08–6.09)	
	HFD + IF	4.34 (2.97–6.13)	
	HFD + E	5.37 (4.09–7.62)	
	HFD + IF + E	4.39 (2.53–5.42)	
Basophils/mm^3^	CG	0.01 (0.01–0.02)	0.723
	CG + IF	0.01 (0.00–0.04)	
	CG + E	0.02 (0.01–0.07)	
	CG + IF + E	0.02 (0.00–0.05)	
	HFD	0.03 (0.00–0.08)	
	HFD + IF	0.03 (0.02–0.05)	
	HFD + E	0.02 (0.01–0.06)	
	HFD + IF + E	0.03 (0.01–0.07)	
T-helper(CD^4^)/mm^3^	CG	0.49 (0.40–0.59) bcd	0.001 *
	CG + IF	0.15 (0.03–0.26) d	
	CG + E	0.57 (0.36–0.84) abcd	
	CG + IF + E	0.43 (0.23–0.64) cd	
	HFD	0.68 (0.28–1.30) abc	
	HFD + IF	0.85 (0.44–1.27) ab	
	HFD + E	0.96 (0.59–1.16) a	
	HFD + IF + E	1.00 (0.70–1.17) a	

* Significant at *p* < 0.05: non significant at *p* > 0.05. a, b, c and d: shows the difference between groups.

**Table 3 medicina-59-01687-t003:** Mean values of monocytes, B-lymphocyte, and eosinophils, and naturel killer levels in all groups and the mean and standard deviation values of live weight.

	Group	$\bar{X}\;\pm \;Sx$ Median (Min–Max)	*p*
	CG	0.16 (0.13–0.20) bc	0.001 *
	CG + IF	0.16 (0.05–0.30) c	
	CG + E	0.27 (0.16–0.42) abc	
B-Lymphocyte /mm^3^	CG + IF + E	0.14 (0.08–0.20) c	
	HFD	0.18 (0.10–0.41) abc	
	HFD + IF	0.28 (0.10–0.30) abc	
	HFD + E	0.30 (0.20–0.41) ab	
	HFD + IF + E	0.32 (0.22–0.38) a	
	CG	0.23 (0.19–0.30) bcd	0.001 *
	CG + IF	0.15 (0.03–0.26) d	
	CG + E	0.27 (0.16–0.42) cd	
Monocytes/mm^3^	CG + IF + E	0.19 (0.11–0.32) abcd	
	HFD	0.29 (0.14–0.57) abcd	
	HFD + IF	0.39 (0.28–0.64) abc	
	HFD + E	0.42 (0.27–0.60) ab	
	HFD + IF + E	0.46 (0.28–0.59) a	
	CG	0.02 ± 0.03	0.169
	CG + IF	0.01 ± 0.05	
	CG + E	0.04 ± 0.15	
Eosinophils/mm^3^	CG + IF + E	0.02 ± 0.04	
	HFD	0.02 ± 0.01	
	HFD + IF	0.03 ± 0.21	
	HFD + E	0.00 ± 0.05	
	HFD + IF + E	0.00 ± 0.00	
	CG	0.23 ± 0.03 b	0.001 *
	CG + IF	0.18 ± 0.12 b	
	CG + E	0.25 ± 0.06 b	
Naturel killer/mm^3^	CG + IF + E	0.20 ± 0.027 b	
	HFD	7.02 ± 3.45 a	
	HFD + IF	0.38 ± 0.10 b	
	HFD + E	0.45 ± 0.09 b	
	HFD + IF + E	0.44 ± 0.07 b	
	CG	270.50 ± 7.92 c	0.001 *
	HFD	385.92 ± 5.27 a	
Live weight (g)	HFD + IF	297.71 ± 5.71 b	
	HFD +E	299.90 ± 5.80 b	
	HFD + IF +E	275.33 ± 7.98 c	

* Significant at *p* < 0.05: non significant at *p* > 0.05. a, b, c and d: shows the difference between groups.

**Table 4 medicina-59-01687-t004:** Mean values of IGF-1, IL-4, TNF-α, IL-12, IL 10 in all groups.

	Group	$\bar{X}\;\pm \;Sx$	*p*
IGF-1 (pg/mL)	CG	115.598 ± 33.205	0.126
	CG + IF	128.901 ± 37.909	
	CG + E	80.644 ± 38.623	
	CG + IF + E	115.962 ± 37.044	
	HFD	76.563 ± 33.614	
	HFD + IF	104.141 ± 21.587	
	HFD + E	97.290 ± 57.474	
	HFD + IF + E	170.503 ± 70.075	
TNF-α (pg/mL)	CG	14.633 ± 4.966 c	0.012 *
	CG + IF	17.817 ± 4.042 bc	
	CG + E	22.250 (12.900–24.300) bc	
	CG + IF + E	21.533 ± 9.369 b	
	HFD	28.200 ± 4.525 a	
	HFD + IF	22.017 ± 3.036 b	
	HFD + E	17.867 ± 4.622 bc	
	HFD + IF + E	18.217 ± 1.747 bc	
IL-4 (pg/mL)	CG	31.769 ± 18.280	0.473
	CG + IF	22.165 ± 9.819	
	CG + E	23.392 ± 11.657	
	CG + IF + E	37.923 ± 13.686	
	HFD	29.578 ± 8.051	
	HFD + IF	25.854 ± 12.010	
	HFD + E	24.892 ± 9.565	
	HFD + IF + E	25.467 ± 7.925	
IL-12 (pg/mL)	CG	2.181 ± 1.781	0.447
	CG + IF	1.948 ± 0.230	
	CG + E	2.432 ± 1.394	
	CG + IF + E	2.282 ± 0.737	
	HFD	2.282 ± 1.048	
	HFD + IF	2.086 ± 1.323	
	HFD + E	1.989 ± 0.786	
	HFD + IF + E	2.853 ± 1.696	
IL-10 (pg/mL)	CG	151.307 ± 19.408 e	0.001 *
	CG + IF	200.495 ± 19.416 d	
	CG + E	194.117 ± 27.545 d	
	CG + IF + E	231.958 ± 37.695 c	
	HFD	240.428 ± 13.687 c	
	HFD + IF	330.343 ± 21.373 b	
	HFD + E	336.845 ± 18.980 b	
	HFD + IF + E	402.237 ± 17.589 a	
IL-1β (pg/mL)	CG	573.107 ± 77.924	0.211
	CG + IF	490.470 ± 90.677	
	CG + E	498.572 ± 213.88	
	CG + IF + E	410.527 ± 149.49	
	HFD	515.736 ± 100.42	
	HFD + IF	412.719 ± 137.27	
	HFD + E	456.762 ± 147.04	
	HFD + IF + E	557.773 ± 69.516	

* Significant at *p* < 0.05: non significant at *p* > 0.05. a, b, c, d and e: shows the difference between groups.

## Data Availability

The data presented in this study are available within the article.

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
