# Peer review of "The Effect of Moderate-Intensity Physical Exercise on Some Serum Inflammation Markers and the Immune System in Rats Fed Intermittent Fasting with a High-Fat Diet"

_medicina, 2023, doi:10.3390/medicina59091687_

Round 1

Reviewer 1 Report

The Authors investigated the effects of moderate-intensity physical exercise on inflammation markers and the immune system in male Wistar albino rats fed a high-fat diet (HFD) with intermittent fasting. The experiment involved 48 rats divided into 8 groups, each with a specific diet and exercise regimen. After 8 weeks, the study found that rats subjected to both exercise and intermittent fasting showed higher lymphocyte levels and increased counts of B lymphocytes and monocytes compared to specific groups. Notably, the group following a high-fat diet displayed elevated TNF-α levels, while the group combining high-fat diet, intermittent fasting, and exercise showed heightened IL-10 levels. The study concludes that in rats on a high-fat diet, the combination of moderate exercise and intermittent fasting led to increased anti-inflammatory cytokines, specific immune cell counts, and decreased pro-inflammatory cytokines. Further research could illuminate the potential implications for human health.

Comments: 

1. Graphical Representation:

   - Utilize bar graphs to visually depict the significant and non-significant values from the table 1, 2 and 3. Clearly label the groups with their respective abbreviations (e.g., CG, CG+IF, etc.) for the ease of understanding. Replace references to group numbers with their corresponding abbreviations (e.g., CG for group 1).

2. Statistical Significance and Elaboration:

   - Specify that the reported differences are statistically significant (e.g., "lymphocyte levels were significantly higher") to provide clearer context.

   - Elaborate on the magnitude of differences where relevant (e.g., "Lymphocyte levels in group HFD+IF were significantly higher by X% compared to CG").

3. English Editing:

   - Thoroughly proofread the manuscript to correct grammar and syntax errors.

4. Methodology Section:

   - Correct any errors, such as "microns" to "microliters."

   - Mention the software used for flow cytometry analysis and statistical analysis.

   - Include Table 4 within the Materials and Methods section.

   - Specify the ELISA kits used for cytokine analysis.

5. Table 4 Correction:

   - Rectify "FITCH" to "FITC" in Table 4.

6. Discussion Section:

   - Retain a focused discussion, emphasizing potential mechanisms linking exercise and intermittent fasting to immune and inflammatory changes.

   - Address the implications of findings for human health, considering applications such as obesity management and chronic disease prevention.

7. Mechanisms and Relevance: (No experiments are needed)

   - Expand on the potential mechanisms by which exercise and intermittent fasting induce immune and inflammatory changes.

   - Discuss the significance of the findings in the broader context of human health, including possible applications for managing inflammation-related conditions.

8. Result section:

I recommend a different approach to presenting the results. Instead of directly starting the result section with Table 1, it would be more informative to first describe the results in detail. After providing this narrative, consider using bar graphs to visually represent the data, and include the tables as supplementary material with all significant and non-significant values. Describing the results in detail before presenting them visually allows readers to better understand the context and significance of the findings. Additionally, by providing all the data in supplementary tables, you ensure transparency and accessibility for readers who may want to delve deeper into the results. This approach enhances the comprehensibility and utility of the study's findings.

9. Whether the authors measure weight changes in the rats to evaluate the impact of dietary and exercise regimes? Weight measurements are commonly employed in studies involving dietary interventions as they can provide critical insights into how different diets and exercise influence body weight and have an impact on immune cell composition and cytokines values. 

By addressing these suggestions, the manuscript can be refined to present a clearer and more comprehensive overview of the study's methods, results, and implications.

I recommend the thorough proofreading of the manuscript to remove grammatical and syntax errors. 

Author Response

The requested corrections have been made and presented in the attchment, I wish you good work

Reviewer 2 Report

The manuscript of GÜNBATAR et al. is well-written and gives a sound explanation of the study and its results.  This article offers findings applicable to other experts. 

There are a few minor points that require improvements and/or clarification that will help to improve the quality of this article:

Line 22, the word the in "The Lymphocyte.." should not be capital

Line 26, correct the word was with were (the IL 10 levels were higher)

Line 77-79, please add a reference for the guidelines you are referring to

Line 91, delete the word a, as it should be "and blood samples"

Line 137, it should be T-helper (you forgot the - )

Thank you.

Minor editing is needed.

Author Response

The requested corrections have been made and presented in the attachment, I wish you good work

Reviewer 3 Report

The study provided a significant finding in which the effects of physical exercise on biomarkers in HFD rats were represented. I did enjoy it the most on reviewing it. The study design is good as up to eight groups of rats were utilized with various intervention and control groups.

However, some issues were spotted, and here are some minute details for improvement;

line no. 13,'' markirs'' should be replaced with ''markers''

line no. 16 and 18, ''standart'' should be replaced with ''standard''

on table 1 and 2, should the solid line be added between each group of parameters (the same as table no.3)

Ref no. 3, 9, 47, 69, 78, 103. Should these ref. be underlined? Rechecking is needed.

Author Response

(The authors gave the same response as above.)

Round 2

Reviewer 1 Report

1.       Remove Unnecessary "th" Instances:

Eliminate instances of "th" where it's not needed, for instance, on line 150 in the result section, where "HFD+IF+Eth" should be corrected. there are numerous places in the manuscript where this mistake has been made.

2.       Spelling Corrections:

Please thoroughly proofread the manuscript and correct any misspellings. For instance, change "Naturell Killer" to "Natural Killer cells."

3.       Unit Correction:

Please change "microns" to "µl" in the material method section, line 112.

4.       Unnecessary Paragraph Removal:

I didn't request an explanation of "flow cytometry" and "FSC, SSC." Kindly remove the unnecessary paragraph in the material and methods section, lines 104-110.

5.       Software Mention:

Please explicitly state the NAME OF THE SOFTWARE used for data analysis in the flow cytometry experiment.

6.       Missing Y-Axis Labels:

The labels on the Y-axis are missing in Figures 1, 2, and 3. Please add them.

7.       Figure Legend Elaboration:

Provide detailed explanations for what the graphs labeled as A, B, and C represent in the figure legends.

8.       Separate Graphs:

Create separate graphs for neutrophils, eosinophils, and basophils. Also, prepare separate graphs for IL-4, IGF-1, and IL-12.

These revisions will enhance the clarity and accuracy of your manuscript.

Please thoroughly proofread the manuscript and correct any misspellings and syntax errors.

Author Response

(The authors gave the same response as above.)
